

# DNA methylome and transcriptome analysis established a model of four differentially methylated positions (DMPs) as a diagnostic marker in esophageal adenocarcinoma early detection

Weilin Peng[1,2], Guangxu Tu[1,2], Zhenyu Zhao[1,2], Boxue He[1,2], Qidong Cai[1,2], Pengfei Zhang[1,2], Xiong Peng[1,2], Shuai Shi[1,2] and Xiang Wang[1,2]

[1] Hunan Key Laboratory of Early Diagnosis and Precise Treatment of Lung Cancer, The Second Xiangya Hospital of Central South University, Central South University, Changsha, Hunan, China
[2] Department of Thoracic Surgery, The Second Xiangya Hospital of Central South University, Central South University, Changsha, Hunan, China

Corresponding author
Xiang Wang, wangxiang@csu.edu.cn

## ABSTRACT

**Background:** Esophageal carcinogenesis involves in alterations of DNA methylation and gene transcription. This study profiled genomic DNA methylome vs. gene expression using transcriptome data on esophageal adenocarcinoma (EAC) tissues from the online databases in order to identify methylation biomarkers in EAC early diagnosis.

**Materials and Methods:** The DNA methylome and transcriptome data were downloaded from the UCSC Xena, Gene Expression Omnibus (GEO), and The Cancer Genome Atlas (TCGA) databases and then bioinformatically analyzed for the differentially methylated positions (DMPs) vs. gene expression between EAC and normal tissues. The highly methylated DMPs vs. reduced gene expression in EAC were selected and then stratified with those of the corresponding normal blood samples and other common human cancers to construct an EAC-specific diagnostic model. The usefulness of this model was further verified in other three GEO datasets of EAC tissues.

**Result:** A total of 841 DMPs were associated with expression of 320 genes, some of which were aberrantly methylated in EAC tissues. Further analysis showed that four (cg07589773, cg10474350, cg13011388 and cg15208375 mapped to gene IKZF1, HOXA7, EFS and TSHZ3, respectively) of these 841 DMPs could form and establish a diagnostic model after stratified them with the corresponding normal blood samples and other common human cancers. The data were further validated in other three GEO datasets on EAC tissues in early EAC diagnosis.

**Conclusion:** This study revealed a diagnostic model of four genes methylation to diagnose EAC early. Further study will confirm the usefulness of this model in a prospective EAC cases.

## INTRODUCTION

Esophageal cancer (EC) is one of the most commonly diagnosed digestive tract carcinomas, ranking as the ninth global cancer burden among all human cancers (*Fitzmaurice et al., 2015*). Histologically, EC can be mainly classified into esophageal squamous cell carcinoma (ESCC) and esophageal adenocarcinoma (EAC), which have distinct epidemiological and pathological characteristics (*Lagergren et al., 2017*; *Xu, 2009*) and different molecular profiles and risk factors (*Lagergren et al., 2017*). For example, ESCC incidence is declined recently, whereas EAC incidence has been increasing in the past decades (*Cook, Chow & Devesa, 2009*; *Edgren et al., 2013*; *Lin et al., 2013*). In treatment of early staged EC, specifically when tumor lesion is limited in the mucosa with no metastasis, either endoscopic mucosal resection (EMR) or dissection together with radiofrequency ablation is the first-line therapy of choice, instead of more aggressive esophagectomy, because such treatment selection is safer, less tissue damages, and largely improved the quality of life for patients as well as better long-term outcomes than those of esophagectomy (*Fitzgerald et al., 2014*; *Lordick et al., 2016*; *Pech et al., 2014*). However, such early staged EC were rarely caught clinically; thus, search for and identification of novel biomarkers could lead to more diagnosis of the early staged ECs. To date, the gold standard in EC diagnosis is endoscopy, but it requires more expense and expertise, and may cause damage to patients (*Smyth et al., 2017*). In this regard, development and evaluation of novel, safer, economic, and effective alternatives in diagnosis of early esophageal cancers are of great clinical significance (*Lagergren et al., 2017*; *Xu, 2009*).

Towards this end, assessment of epigenetic alterations in various human cancers showed a great potential as biomarkers in cancer early diagnosis; for example, detection of aberrant DNA methylation, one of the major forms in epigenetic alterations, had also been observed to associate with development of numerous human diseases, including cancer (*Robertson, 2005*). Altered DNA methylation showed as an early step in transformation of metaplasia to dysplasia and neoplasia (*Alvarez et al., 2011*; *Kaz et al., 2011*; *Smith et al., 2008*; *Xu et al., 2013*). DNA methylation, a process in transferring a methyl group onto the C5 position of the cytosine, can negatively regulate gene expression in general; however, aberrantly hypermethylation of tumor suppressor genes could silence expression of these genes and promotes or contributes to cancer development (*Hao et al., 2017*). For example, methylation of Ras association domain-containing protein 1A (*RASSF1A*) occurred in ovarian cancer (*Si et al., 2014*) and esophageal cancer (*Guo et al., 2016*; *Yang et al., 2014*). Moreover, alterations of DNA methylation (*Laird, 2003*) and tumor-specific DNA methylation profiles could be detected repeatedly in early stage of cancers (*Koch et al., 2018*). In ESCC, aberrantly methylated genes were significantly enriched in IL-10 anti-inflammatory signaling and cell communication pathway (*Lima et al., 2011*). Different from ESCC, several cancer-associated pathway genes were aberrantly methylated in EAC, including genes in the epithelial-mesenchymal transition (EMT), cell adhesion, Wingless and Int-1 (WNT), and Transforming growth factor (TGF) pathways (*Kaz et al., 2011*; *Xu et al., 2013*). However, to date, most of these DNA methylome studies of EAC were restricted to handful CpG island sites of selected genes,
like the Illumine Human Methylation 27K array (*Alvarez et al., 2011*; *Kaz et al., 2011*; *Smith et al., 2008*; *Xu et al., 2013*).

In this study, we conducted a systematically investigation into EAC genome-wide methylome and transcriptome profiles to identify genome-wide differentially hypermethylated CpG loci using the online The Cancer Genome Atlas (TCGA) and the Gene Expression Omnibus (GEO) databases. Afterwards, we identified different DNA methylation sites in EAC tissues vs. those of normal esophagus tissues and then selected those with reduced expression of the corresponding genes, leading to four hypermethylation sites by using the least absolute shrinkage and selection operator (LASSO) regression analysis and to establishing a model of these four sites as a diagnostic model using the logistic regression analysis. We then verified this model in other three independent datasets (GSE81334, GSE89181 and GSE104707). We expected to provide a useful model in help in early EAC diagnosis.

## MATERIALS AND METHODS

### Database search and data download and initial analysis

We first performed database search and downloaded data from UCSC Xena, TCGA, and GEO databases, respectively. Our detailed data process and analysis procedures are listed in Fig. 1. In brief, DNA methylation data on EAC and normal squamous esophagus were collected from the UCSC Xena database (https://xenabrowser.net/) (*Goldman et al., 2015*). The database includes DNA methylation data in different human cancers, while there were two counts in a given CpG island, i.e., a methylated intensity (M) and an unmethylated intensity (U) and the methylation levels (β) were summarized as M/(M+U+100). Clinicopathological data and RNA-Seq data on gene expression in EAC were downloaded from TCGA (https://portal.gdc.cancer.gov/). The DNA methylation data on other human cancers were also extracted from the UCSC Xena database, including ACC_T, Adrenocortical carcinoma; BLCA_T, Bladder Urothelial carcinoma; BRCA_T, Breast invasive carcinoma; CESC_T, Cervical squamous cell carcinoma and endocervical adenocarcinoma; CHOL_T, Cholangiocarcinoma; COAD_T, Colon adenocarcinoma; DLBC_T, Diffuse Large B-cell Lymphoma; GBM_T, Glioblastoma multiforme; HNSC_T, Head and Neck squamous cell carcinoma; KICH_T, Kidney Chromophobe; KIRC_T, Kidney renal clear cell carcinoma; KIRP_T, Kidney renal papillary cell carcinoma; LAML_T, Acute Myeloid Leukemia; LGG_T, Brain Lower Grade Glioma; LIHC_T, Liver hepatocellular carcinoma; LUAD_T, Lung adenocarcinoma; LUSC_T, Lung squamous cell carcinoma; MESO_T, Mesothlioma, OV_T, Ovarian serous cystadenocarcinoma; PAAD_T, Pancreatic adenocarcinoma; PCPG_T, Pheochromocytoma and Paraganglioma; PRAD_T, Prostate adenocarcinoma; READ_T, Rectum adenocarcinoma; STAD_T, stomach adenocarcinoma; UCEC_T, Uterine Corpus Endometrial Carcinoma. In addition, we also downloaded five independent DNA methylation datasets on EAC and normal squamous esophagus from the GEO database, including GSE72872 (*Krause et al., 2016*), GSE69270 (*Kananen et al., 2016*; *Mishra et al., 2020*), GSE81334, GSE89181 (*Kaz et al., 2016*), and GSE104707 (*Luebeck et al., 2017*). These data were then processed by using the Illumina 450K array, analyzed, and annotated by the ChAMP R software package (*Tian*

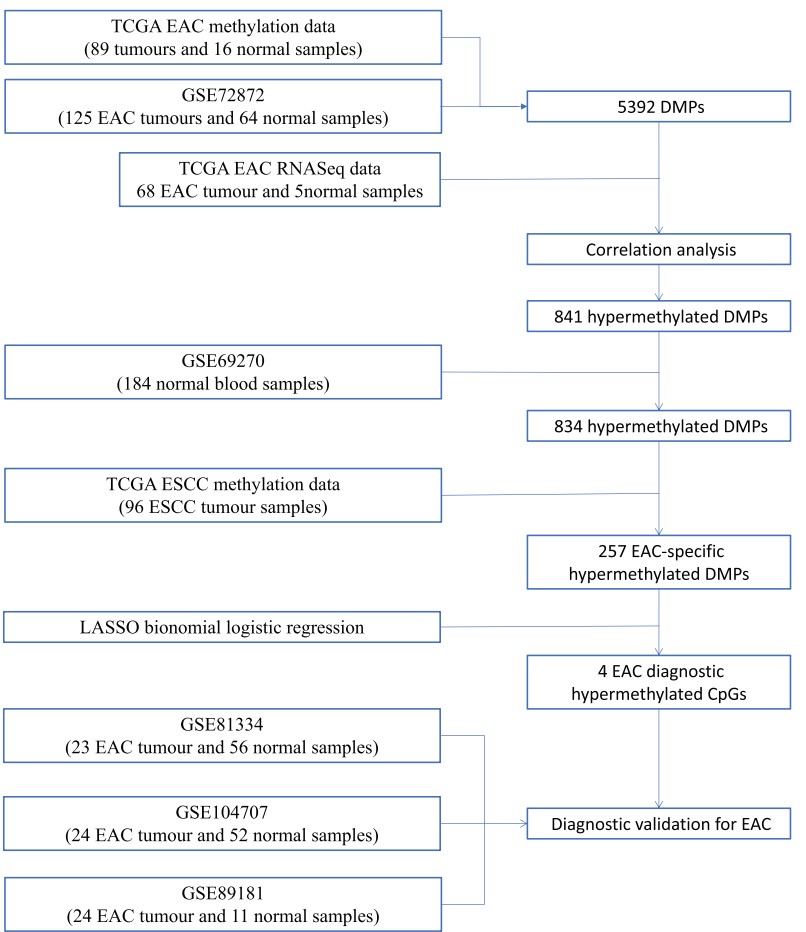

**Figure 1 Workflow of the study.** Illustration of data process and analyses of this study.

*et al., 2017*). However, the samples without tumor stage information were removed, the CpG methylation data missing in more than 10% of samples were also filtered out, and data using the probes being localized in allosome, non-CpG probes or single-nucleotide polymorphism-related probes, or multi-hit probes were further removed by the champ. filter function of the ChAMP R package. After that, the rest real CpG probes with no methylation values were processed using the R impute.knn function. In addition, the RNA-Seq data on gene expression in EAC tissues were acquired from TCGA dataset, and subsequently analyzed for differentially expressed genes vs. the normal esophageal tissues using DESeq R package.

## Analysis of differential DNA methylation CpGs vs. differentially expressed genes

To identify the differentially methylated positions (DMPs) involved in the early EAC tumorigenesis, we assigned TCGA EAC tumor data into two subtypes stratified by stage I/II vs. III/IV. Since gene transcription was significantly influenced by level of the CpG island methylation in their gene promoter regions, the CpG sites localized at the 200 bp

upstream of the transcription start site (TSS200) and/or TSS1500 were selected for further analysis. Significant DMPs were identified by using the functional champ.DMP in the ChAMP R package, while the criterion of DMPs was set with the false discovery rate (FDR) less than 0.05, while the absolute difference in β value exceeded 0.2. Thus, the CpG sites met this criterion were considered as significant DMPs. After that, we selected the intersectional DMPs on TCGA EAC and GEO72872 datasets as candidate DMPs.

## Data correlational analysis and construction of a candidate diagnostic DNA methylation model

We performed Spearman correlation test and selected DMPs, whose level were significantly negatively associated with gene expression using a *P* value less than 0.05 and coefficient less than −0.4. We then identified and selected candidate DMPs as a diagnostic model, i.e., we first searched the differentially methylated CpGs by analyzed them for three times, i.e. TCGA stage I/II EAC samples vs. the normal squamous esophagus samples, TCGA stage III/IV EAC samples vs. normal squamous esophagus samples, and GSE72872EAC samples vs. the normal squamous esophagus samples. We identified a total of 4852 hypermethylated CpG sites in all three comparisons and then associated them with expression of their corresponding genes and found 841 hypermethylated CpG sites to negatively associate with gene expression. Furthermore, we stratified these 841 DMPs with that of normal blood samples of GSE69270 dataset to exclude seven of them based on a higher average β value than that of EAC samples, leading to the remaining 834 DMPs. After analyzed them against the ESCC samples, we obtained 257 DMPs with a mean methylation level less than 0.1 in ESCC samples and then utilized dichotomous logistic regression analysis and LASSO methods to obtain four DMPs as a model to diagnose EAC early.

## Validation of the diagnostic model of the four DMPs in EAC

After that, we verified the usefulness of this model in EAC diagnosis based on the β value and risk score between 24 EAC samples and 11 normal squamous esophagus samples in GSE89181 dataset. We then constructed the receiver operating characteristics (ROC) using the pROC R package according to a previous study (*Robin et al., 2011*) and obtained the area under the curve (AUC) value for the specificity and sensitivity. Afterwards, we performed the same analysis for the GSE72872, GSE81334 and GSE104707 data. And usefulness of this model in early staged EAC diagnosis was also verified in stage I EAC cancers of TCGA and GSE89181 by using the same method.

## Statistical analysis

We performed Shapiro–Wilk test to normalize the downloaded data and Wilcoxon rank-sum test to compare methylation levels and the risk scores in different data groups, e.g., EAC vs. normal tissues. We also conducted the Spearman correlation test to correlate methylation levels and gene expression. All statistical analyses were performed by using the R 3.6.2 (www.r-project.org) or GraphPad Prism 8.2.1 (GraphPad Software, La Jolla,

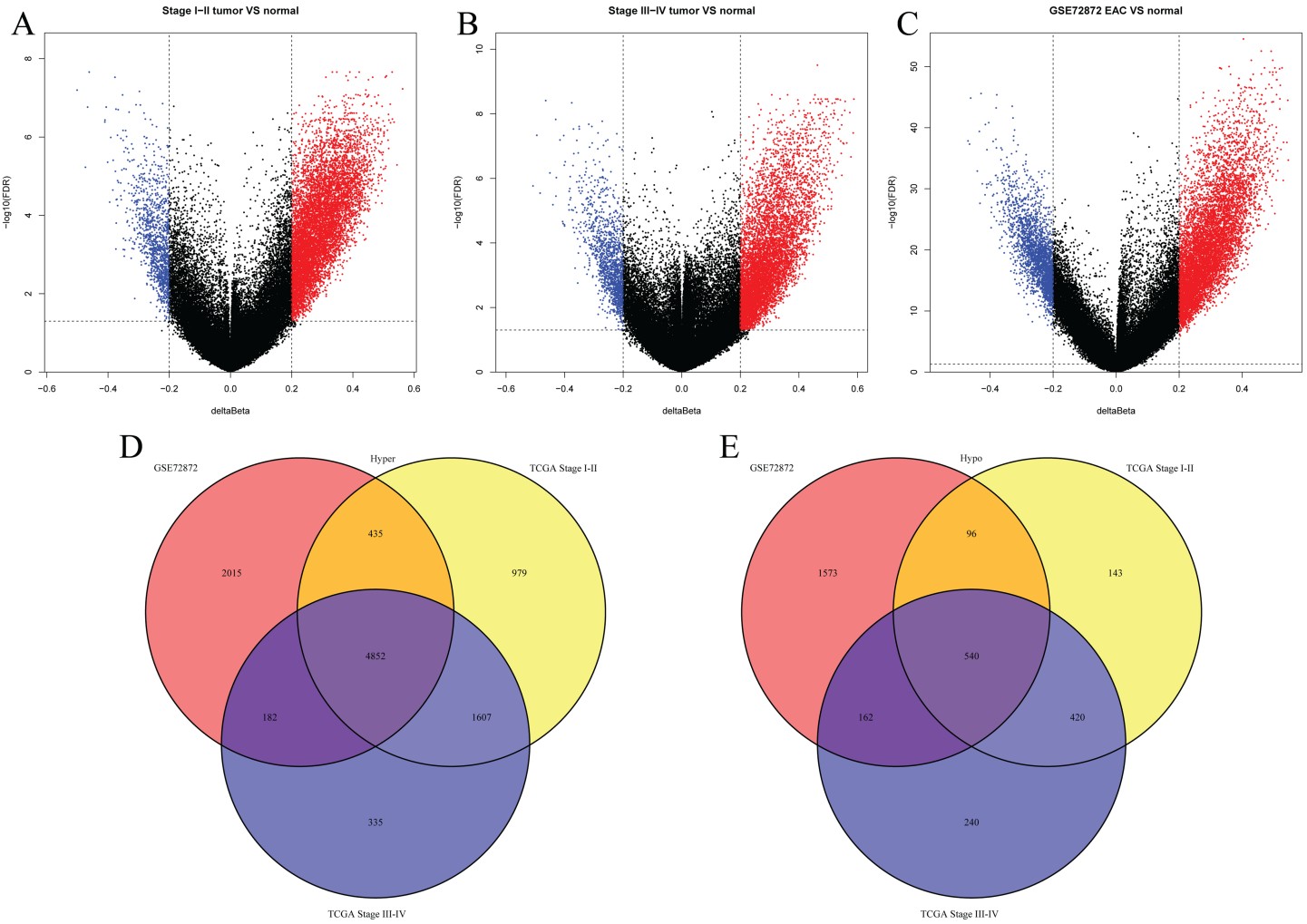

**Figure 2 Differentially methylated CpG sites using TCGA EAC and GSE72872 datasets.** (A–C) The volcano map of DMPs between (A) TCGA stage I/II EAC, (B) TCGA stage III/IV EAC, (C) EAC in GSE72872 vs. the normal samples. (D and E) The Venn diagrams. The data show the coincidence of 4,852 hypermethylated DMPs (D) and 540 hypomethylated DMPs (E) in TCGA EAC and GSE72872 datasets.

CA, USA). A p value that was equal to or less than 0.05 was considered as statistically significant.

# RESULTS

## Identification of DMPs in EAC

We first analyzed DMPs using data on 89 EAC and 16 normal squamous esophagus samples from the UCSC Xena dataset, while EAC samples without TNM information were excluded, leading to 34 stage I/II and 34 stage III/IV samples. We found a total of 95248 CpG sites localized at the 200 bp upstream of the transcription start site (TSS200) or TSS1500 and 9072 DMPs between stage I/II EAC and normal samples (Fig. 2A) and 8338 DMPs between stage III/IV EAC and normal esophagus tissues (Fig. 2B). Similarly, we found 9855 DMPs out of 91602 methylated CpG sites in 125 EAC and 64 normal samples

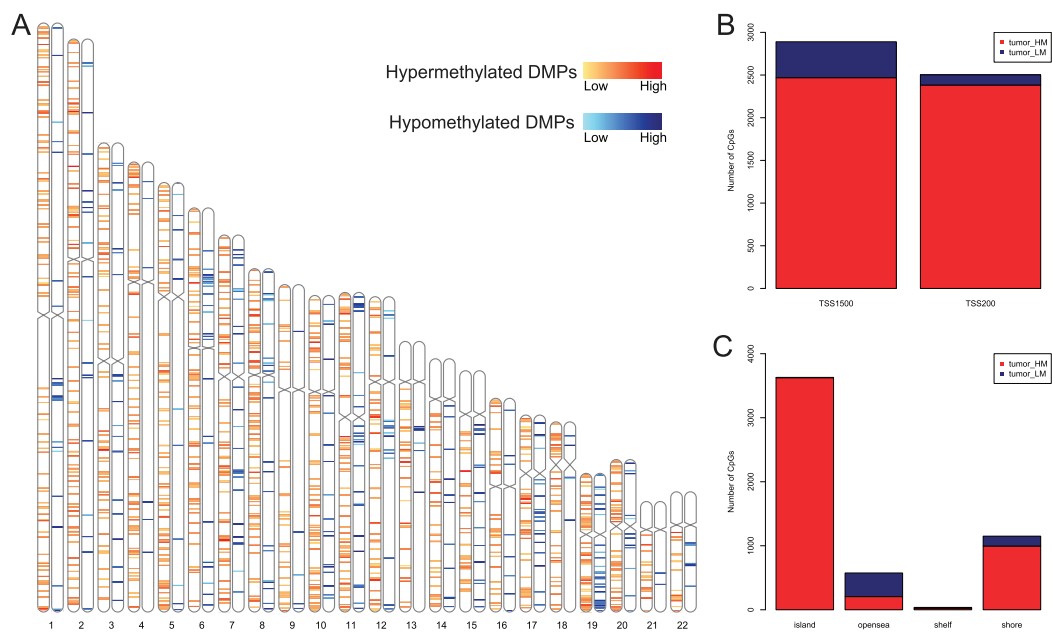

**Figure 3 Distribution of differentially methylated positions (DMPs).** (A) Distribution of the 5392 DMPs in different chromosomes. (B) Distribution of the 5392 DMPs in the 1,500 bp and/or 200 bp upstream of the transcription starting sites (TSS1500 and/or TSS200). (C) Distribution of the 5392 DMPs in terms of CpG features.                               

from GSE72872 dataset (Fig. 2C). Ultimately, we identified a total of 5392 DMPS (4852 hypermethylated and 540 hypomethylated DMPs) coinciding in all three cohorts (Figs. 2D–2E). We then performed the GO and KEGG pathway analyses of the corresponding 1796 genes and the data are shown in Fig. S1 i.e., these genes were significantly enriched in cell growth, EMT, PI3K-Akt signaling and Wnt signaling pathway, respectively. These DMPs localized in the different chromosomes (Fig. 3A) and the quantity of these DMPs showed that they were more concentrated in chromosome 1 and 2 with 384 and 409 DMPs, respectively, whereas chromosome 21 and 22 had least DMPs with 58 and 52, respectively. The majority of these DMPs were hypermethylated (90%), although their distribution had no distinct difference in TSS200 or TSS1500 (Fig. 3B). The number of the hypermethylated DMPs was reduced dramatically in "opensea", "shelves" and "shores" regions (Fig. 3C).

## Association of DMPs with expression of the corresponding genes

Of these 5392 DMPs coinciding in all three cohorts of EAC samples, the 4527 hypermethylated DMPs were corresponding to 1,262 genes and 475 hypomethylated DMPs to 374 genes. Moreover, 841 hypermethylated DMPs was reversely associated expression of 320 genes and 57 hypomethylated DMPs were negatively associated with expression of 43 genes based on Spearman correlation analysis of methylome and transcriptome data. The heatmap of these 898 DMPs is illustrated in Fig. 4A, while the heatmap of the corresponding genes is shown in Fig. 4B, suggesting that there was a
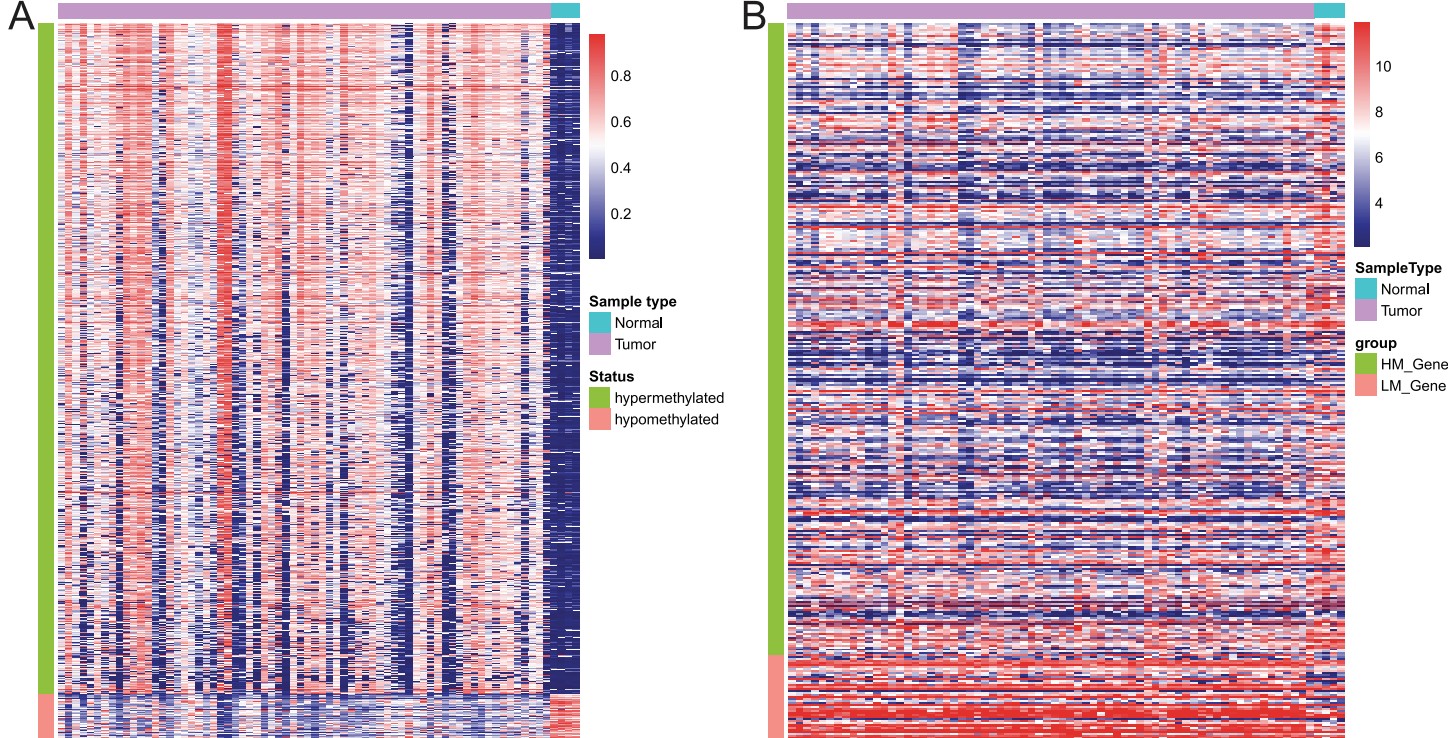

**Figure 4 Heatmap of differentially methylated CpGs and expression of corresponding genes.** (A) Heatmap of methylation levels of the 898 differentially methylated positions. (B) Heatmap of expression of the 363 differentially methylated genes. HM_gene, hypermethylated gene, LM_gene, hypomethylated gene.

distinct difference in methylation level and gene expression between EAC and normal samples.

## Establishment of DMPs as a diagnostic model in EAC

After that, we screened hypermethylated DMPs between EAC and normal samples and excluded those with higher methylation levels in normal blood samples than in EAC tissues. We found a total of 834 hypermethylated DMPs, among which 649 CpGs also occurred in all six datasets, and heatmap of these 649 CpGs is shown in Fig. 5A. Our unsupervised hierarchical clustering analysis revealed that EAC samples had distinct difference in DMPS levels vs. those of normal blood or normal esophagus samples. Furthermore, we excluded 577 from 834 DMPs based on the mean β value greater than 0.1 in ESSC samples, leading to 257 DMPs to be able to diagnose EAC. We then performed the binary logistic regression and LASSO methods and identified four CpGs to construct the risk score model (Table 1), in which we utilized the formula: The risk score = 2.186 × β value of cg07589773 + 0.504 × β value of cg10474350 + 1.550 × β value of cg13011388 + 2.371 × β value of cg15208375. The risk score of different EAC datasets was significantly higher than that of the ESSC dataset (Fig. 5B), while the methylation levels of these four CpGs were compared in all other cancer types from the UCSC Xena database (Fig. 5C). These results showed that these four CpGs were the robust candidate EAC-specific diagnostic biomarkers. These four CpGs were mapped to four different genes

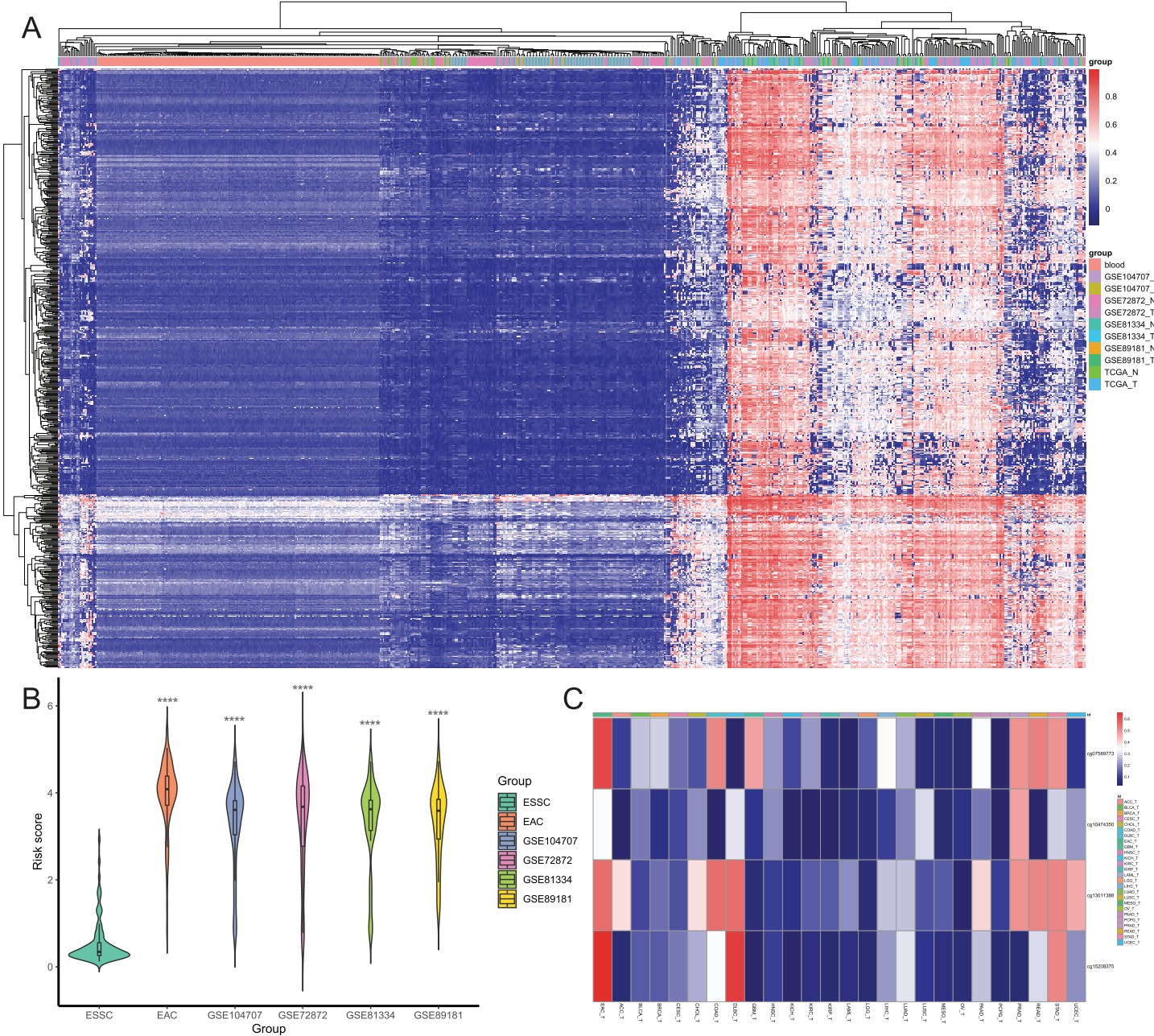

**Figure 5** **Screening of candidate EAC-specific diagnostic biomarkers.** (A) Heatmap of methylation levels of the 649 EAC hypermethylated CpGs among the blood, different EAC and normal samples. (B) The risk score. The score was calculated using the diagnostic model of each ESSC and different EAC datasets. (C) Heatmap of the average methylation levels of four candidate CpGs in EAC and other cancer types. ACC_T, Adreno-cortical carcinoma; BLCA_T, Bladder Urothelial carcinoma; BRCA_T, Breast invasive carcinoma; CESC_T, Cervical squamous cell carcinoma and endocervical adenocarcinoma; CHOL_T, Cholangiocarcinoma; COAD_T, Colon adenocarcinoma; DLBC_T, Diffuse Large B-cell Lymphoma; GBM_T, Glioblastoma multiforme; HNSC_T, Head and Neck squamous cell carcinoma; KICH_T, Kidney Chromophobe; KIRC_T, Kidney renal clear cell carcinoma; KIRP_T, Kidney renal papillary cell carcinoma; LAML_T, Acute Myeloid Leukemia; LGG_T, Brain Lower Grade Glioma; LIHC_T, Liver hepatocellular carcinoma; LUAD_T, Lung adenocarcinoma; LUSC_T, Lung squamous cell carcinoma; MESO_T, Mesothlioma, OV_T, Ovarian serous cystadenocarcinoma; PAAD_T, Pancreatic adenocarcinoma; PCPG_T, Pheochromocytoma and Paraganglioma; PRAD_T, Prostate adenocarcinoma; READ_T, Rectum adenocarcinoma; STAD_T, stomach adenocarcinoma; UCEC_T, Uterine Corpus Endometrial Carcinoma.

**Table 1 Information of the four candidate CpGs.**

| CpG | Chromosome | Gene | Feature | Start | End | Coefficient |
|-----|-----------|------|---------|-------|-----|-------------|
| cg07589773 | chr7 | IKZF1 | TSS1500-island | 50304287 | 50304288 | 2.186 |
| cg10474350 | chr7 | HOXA7 | TSS1500-island | 27156936 | 27156937 | 0.504 |
| cg13011388 | chr14 | EFS | TSS200-island | 23365700 | 23365701 | 1.55 |
| cg15208375 | chr19 | TSHZ3 | TSS1500-island | 31350649 | 31350650 | 2.371 |

(IKZF1, HOXA7, EFS, and TSHZ3), and our Kaplan Meier curves of these four genes showed that their expression level did not affect prognosis of EAC patients (Fig. S2). The process of the CpG methylation was regulated by many factors, some of which had been identified, including DNA methyltransferase, DNA demethylase, histone methyltransferase, and histone demethylase. The correlation between these regulators and these four genes were investigated, results of which showed that IKZF1 was significantly associated with SETBP1 level and inversely associated with levels of SUV39H1, SET, SMYD5, SUV39H2, DPY30 and WDR5; EFS significantly associated with PRDM2, SETBP1 and TET2 but inversely associated with DNMT1; TSHZ3 significantly associated with SETBP1 and inversely associated with SUV39H1, SET, SMYD5, SUV39H2, IDH2, and WDR5 (Fig. S3).

## Validation of this four DMPS as a signature to diagnose EAC

After that, we verified the usefulness of this model of four DMPs as diagnostic markers for EAC. Figs. 6A–6D illustrated that there was significant higher methylation levels of four DMPs between tumor and normal samples in external datasets. Since methylation in adjacent sites had similar pattern (*Bibikova et al., 2011*), methylation levels of other CpG sites in TSS200 and/or TSS1500 of four genes mapped to these four CpGs were examined, result showed all other CpG sites were hypermethylated in EAC tissues vs. that of normal esophagus tissues in TCGA EAC dataset (Figs. 7A–7D), consistent with the four CpGs. The risk scores calculated based on the model also showed significant difference between EAC and normal samples (Figs. 8A–8D and Fig. S4A). The AUC values of this model in TCGA, GSE72872, GSE81334, GSE89181 and GSE104707 datasets were 0.9, 0.95, 0.989, 0.992 and 0.998, respectively (Figs. 9A–9D and Fig. S4B). The sensitivity and specificity of this model in EAC diagnosis were 0.978 and 0.875, 0.88 and 0.953, 1 and 0.893, 0.917 and 1, 1 and 0.962, respectively in TCGA, GSE72872, GSE81334, GSE89181 and GSE104707 datasets. Furthermore, we assessed the usefulness of this model of the four methylation biomarkers in diagnosis of early EAC. We first assessed the methylation levels of the four CpGs between stage I EAC and normal tissues in GSE89181 and the result showed all four CpGs were hypermethylated in stage I EAC vs. normal esophagus (Fig. S5). The risk score of the stage I EAC vs. the normal tissues in TCGA and GSE89181 datasets was also distinctly different (Figs. 10A, 10C). The ROC curves were plotted for TCGA and GSE89181 stage I EAC (Figs. 10B, 10D). In TCGA dataset the AUC value was 0.903 and the sensitivity and specificity were 1 and 0.812,

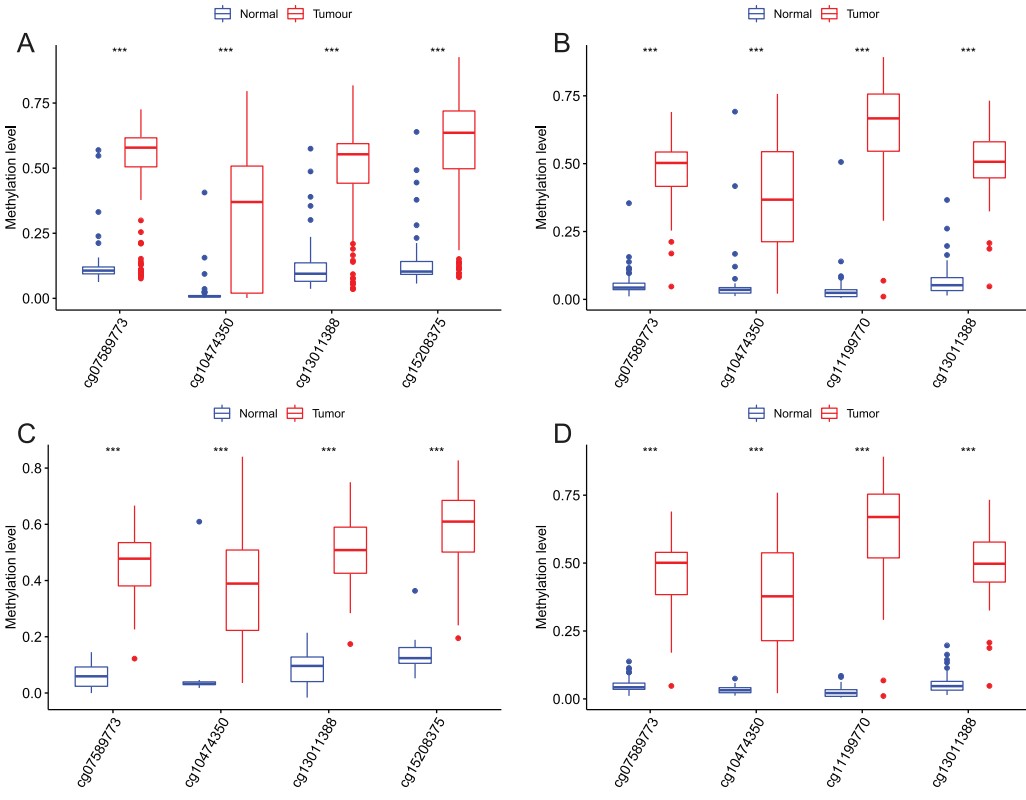

**Figure 6** **Methylation levels of these four CpGs between EAC and normal samples in other EAC datasets.** Four CpGs methylation levels between EAC and normal samples in GSE72872 (A), GSE81334 (B), GSE89181 (C), and GSE104707 (D).

respectively. The AUC value and the sensitivity and specificity were all 1 in GSE89181 dataset.

## DISCUSSION

In the current study, we first profiled global DNA methylations on EAC identified by using the Illumina Human Methylation 450K BeadChips and then compared the data with gene expression data on RNA-Seq datasets. We identified differential methylated CpGs and stratified by their gene expression and found a total of 841 hypermethylated DMPs with downregulated genes. After that, we further stratified these 841 DMPs with those of ESCC and normal blood samples, and performed binary logistic regression and LASSO analyses to obtain four of them as a model in early diagnosis of EAC. These four DMPs corresponding to four different protein-coding genes, i.e., Ikaros family zinc finger protein 1 (IKZF1), Homeobox protein Hox-A7 (HOXA7), Embryonal fyn-associated substrate (EFS), and Teashirt homolog 3 (TSHZ3). We then validated this model of the four methylation biomarkers in other three independent GEO datasets. Our model also showed very high values of the AUC and high sensitivity and specificity in early EAC diagnosis. Indeed, the characteristics of a clinical applicable diagnostic biomarker should include high sensitivity and specificity in diagnosis of early stage cancer, but not mistakenly diagnose other cancer types or non-cancerous diseases, in addition to non-invasive and

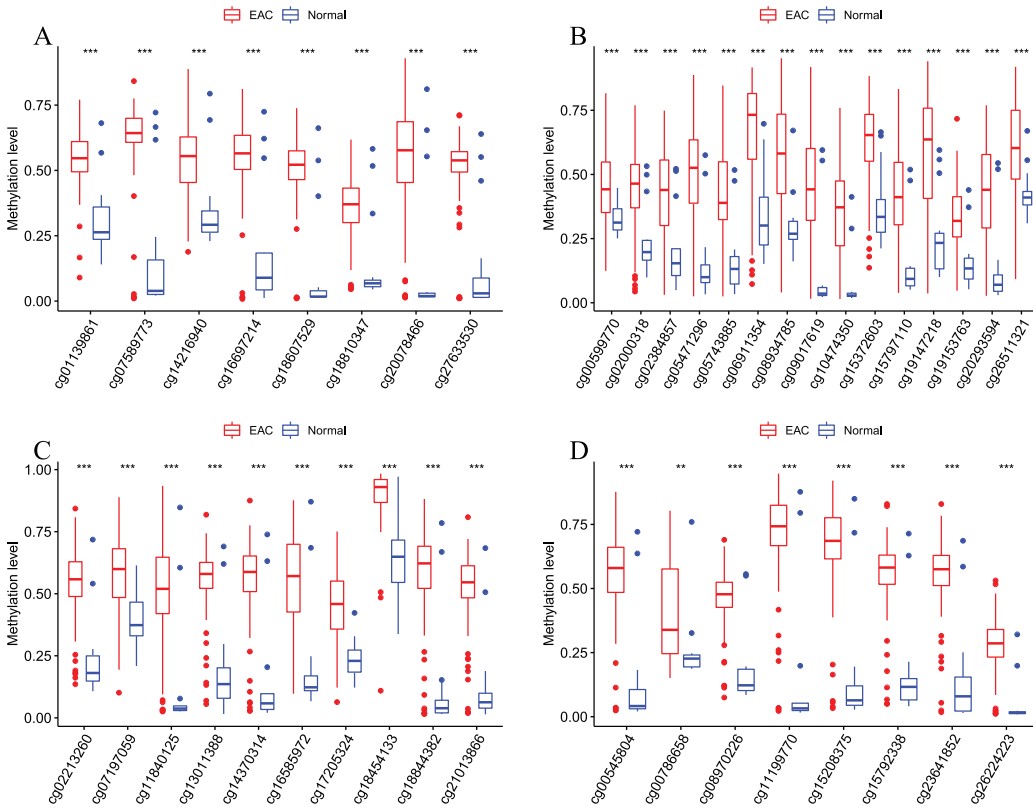

**Figure 7 Methylation levels of all CpGs at the promoter regions of these four corresponding genes.**
Methylation levels of the promoter CpG islands of these four corresponding genes IKZF1(A), HOXA7
(B), EFS(C), and TSHZ3 (D) between EAC and normal samples in TCGA. **$p < 0.01$ and ***$p < 0.001$.

relatively economic procedure. In the present study, we established this model of four
CpGs for EAC diagnosis, indicating a great potential biomarker in early EAC detection,
although this model was not validated in the blood or other non-invasive biospecimens.
Future study will evaluate it in serum samples to verify the usefulness.

DNA methylation epigenetically modifies gene expression; thus, aberrant gene
promoter methylation was associated with cancer development and progression
(*Robertson, 2005*). The current study profiled global DNA methylations on EAC and
identified a large set of genomic DNA methylations on EAC, some of which were shown in
previous studies. For example, the E3 ubiquitin-protein ligase CHFR showed to be
required in maintenance of the antephase checkpoint that regulates cell cycle and was
hypermethylated in EAC (*Soutto et al., 2010*). The promoters of glutathione peroxidase 7
(*GPX7*) and glutathione S-transferase Mu 2 (*GSTM2*) were reported to be frequently
hypermethylated in 67% and 69% of EAC, respectively, expression of which was also
reduced respectively (*Peng et al., 2009*). Moreover, aberrant hypermethylation of the
secreted frizzled-related protein 1 (*SFRP1*) promoter regions associated with reduced
SFRP1 expression, which occurred in early EAC (*Zou et al., 2005a*), while altered
methylation in Eyes absent homolog 4 (*EYA4*) promoter occurred in esophageal mucosa

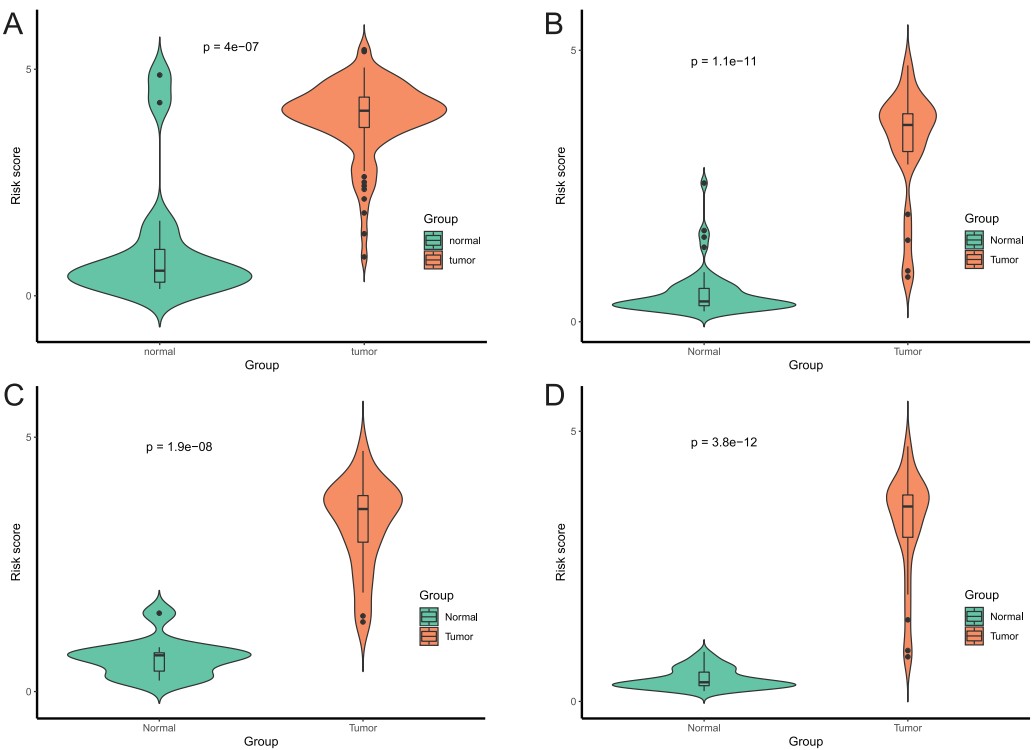

**Figure 8 Evaluation power of this diagnostic model.** The risk score was calculated by using data of the model of EAC and normal samples in TCGA (A), GSE81334 (B), GSE89181 (C), and GSE104707 (D).

metaplasia and Barrett's esophagus progression to EAC (*Zou et al., 2005b*) and hypermethylation of Runt-related transcription factor 3 (*RUNX3*) promoter was reported as an independent risk factor associated with Barrett's esophagus-related EAC development (*Schulmann et al., 2005*). However, detection and analysis of this large amount of gene methylations as biomarker are not practical. Thus, for EAC-specific biomarker development, we further stratified our DNA methylation data using those of normal blood samples and performed LASSO analyses to narrow down to four DMPs, which greatly reduced the numbers of gene methylations and also validated them in other three independent EAC samples and stage I EAC samples. We can therefore confidently suggest that this model of the four DNA methylation markers could be useful in early EAC detection.

As we know, IKZF1, a transcription factor, is a known regulator of immune cells development, mainly in early B cells, CD4+ T cells, altered IKZF1expression has been linked to the development of chronic lymphocytic leukemia (*Kastner & Chan, 2011*; *Oliveira et al., 2019*), while HOXA7 is a DNA-binding transcription factor to regulate gene expression, morphogenesis, and differentiation (*Li, Ye & Zhou, 2020*) and showed upregulated expression in ESCC (*Chen et al., 2005*) and laryngeal squamous cell cancer (*Li, Ye & Zhou, 2020*), whereas it was downregulated expression in renal clear cell carcinoma (*Cui et al., 2020*). Moreover, EFS protein acts as a scaffolding protein for cell

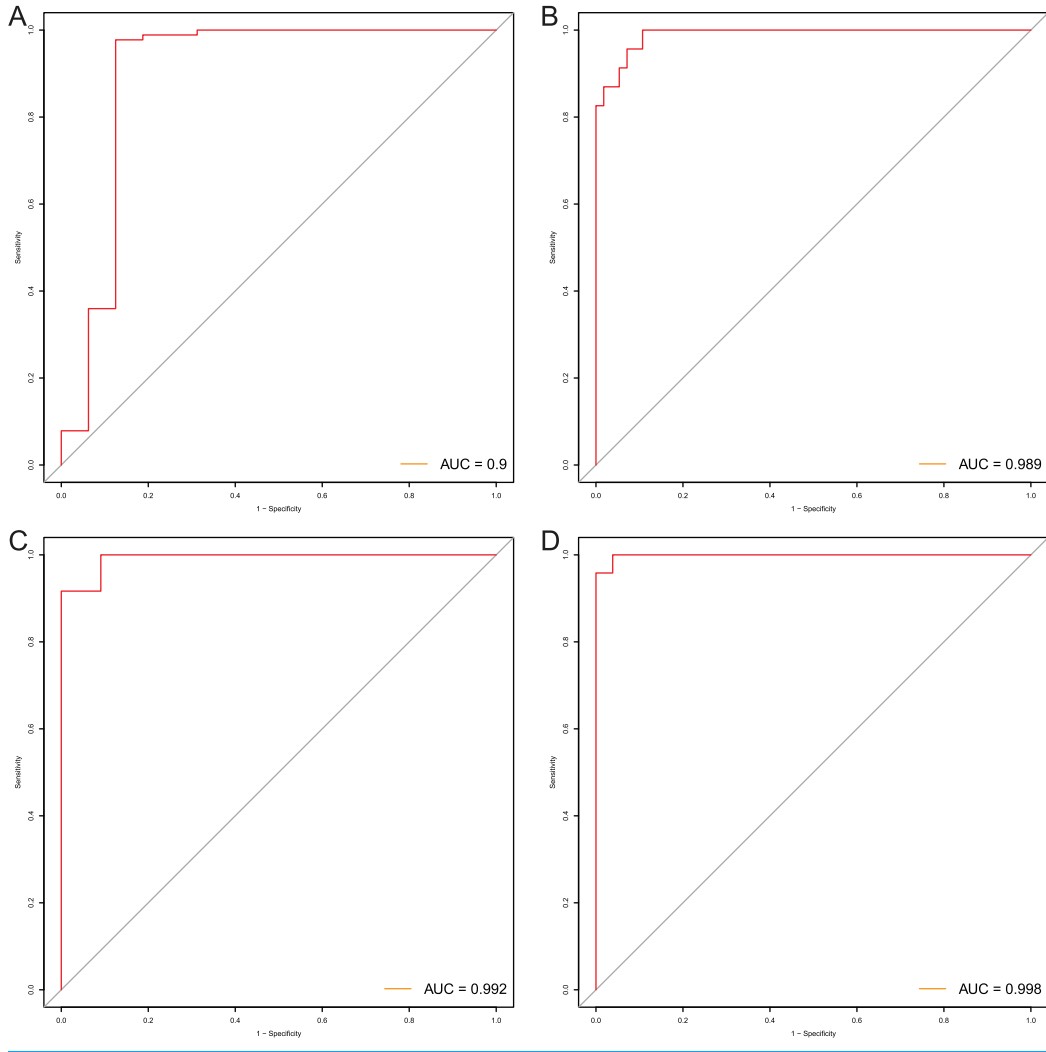

**Figure 9 Evaluation power of this diagnostic model.** The ROC curves of this diagnostic model in TCGA (A), GSE81334 (B), GSE89181 (C), and GSE104707 (D).

signaling in the immune system and altered EFS expression was associated with cancer development (*Neumann et al., 2011*; *Sertkaya et al., 2015*; *Vital et al., 2010*). In addition, TSHZ3 controls breathing and has been identified as a critical region of the heterozygous deletions at 19q12-q13.11, in development of autism spectrum disorder symptoms (*Caubit et al., 2016*). TSHZ3 expression, as one of five genes, was high in tongue SCC (*Zeng et al., 2019*) and TSHZ3 was duplicated in ovarian and triple negative breast cancers (*McBride et al., 2012*), while TSHZ3, as one of ten genes, inhibited the p53 activity in multiple non-small cell lung cancer cell lines (*Van Olst et al., 2017*). However, to date, there are no reports of aberrant expression or activities of these four genes in EAC. The role of these four genes in EAC needs more study.

However, it is the limitation of this study, a technique that can sensitively detect methylation conditions of the circulating tumor DNAs will be more useful; thus, we will investigate whether this model of the four gene methylations can use blood samples.

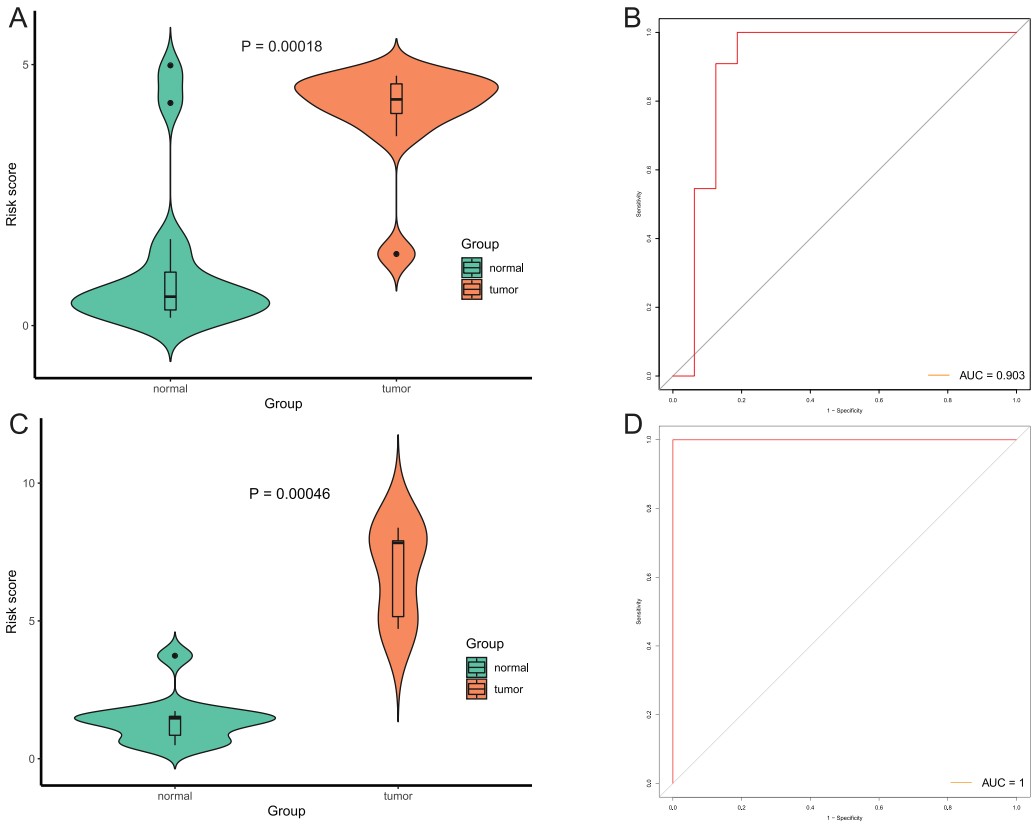

**Figure 10 Evaluation diagnostic performance of this model in diagnosis of early stage EAC.** (A and C) The risk score of this model in stage I EAC and normal samples in TCGA (A) and GSE89181 (C). (B and D) The ROC curves of this model in stage I EAC and normal samples in TCGA (B) and GSE89181 (D).

Although these four methylation sites in EAC had relatively higher β values than those in other cancer types, this model could not effectively distinguish EAC from other cancer types. So identification of more specific methylation sites will be warranted in our future work. Moreover, another limitation of this study was that we didn't further assess the role of these four genes in control of esophageal tumorigenesis.

## CONCLUSION

The current data demonstrated that detection of aberrant DNA methylation in four CpG sites as a model could sensitively and specifically diagnose EAC early in four different online EAC datasets. Further study will validate and investigate the mechanisms underlying aberrant methylation in EAC tumorigenesis. Future study will also validate this promising diagnostic model of four biomarkers in prospective clinical EAC samples.

## ACKNOWLEDGEMENTS

The authors would like to thank Professor Yongguang Tao for his instruction throughout the designing and writing process. It is with regret that not all relevant studies could be cited due to space limitations.

## Funding

This work was supported by the National Natural Science Foundation of China (81672308, 81101767 and 81972195), the Hunan Provincial Key Area R&D Programmes (2019SK2253), and the Scientific Research Program of Hunan Provincial Health Commission (20201047). The funders had no role in study design, data collection and analysis, decision to publish, or preparation of the manuscript.

## Grant Disclosures

The following grant information was disclosed by the authors:
National Natural Science Foundation of China: 81672308, 81101767 and 81972195.
Hunan Provincial Key Area R&D Programmes: 2019SK2253.
Scientific Research Program of Hunan Provincial Health Commission: 20201047.

## Competing Interests

The authors declare that they have no competing interests.

## Author Contributions

- Weilin Peng conceived and designed the experiments, performed the experiments, analyzed the data, prepared figures and/or tables, authored or reviewed drafts of the paper, and approved the final draft.
- Guangxu Tu analyzed the data, prepared figures and/or tables, and approved the final draft.
- Zhenyu Zhao analyzed the data, prepared figures and/or tables, and approved the final draft.
- Boxue He analyzed the data, prepared figures and/or tables, and approved the final draft.
- Qidong Cai performed the experiments, analyzed the data, prepared figures and/or tables, and approved the final draft.
- Pengfei Zhang analyzed the data, prepared figures and/or tables, and approved the final draft.
- Xiong Peng analyzed the data, prepared figures and/or tables, and approved the final draft.
- Shuai Shi analyzed the data, prepared figures and/or tables, and approved the final draft.
- Xiang Wang conceived and designed the experiments, prepared figures and/or tables, and approved the final draft.

## Data Availability

DNA methylation data (Methylation450K) of ESCA, ACC, BLCA, BRCA, CESC, CHOL, COAD, DLBC, GBM, HNSC, KICH, KIRC, KIRP, LAML, LGG, LIHC, LUAD, LUSC, MESO, OV, PAAD, PCPG, PRAD, READ, STAD and UCEC were collected from the UCSC Xena database (https://xenabrowser.net/datapages/).

GSE69270, GSE72872, GSE89181, GSE81334 and GSE104707 datasets are available from GEO.

## Supplemental Information

Supplemental information for this article can be found online at http://dx.doi.org/10.7717/peerj.11355#supplemental-information.

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
