# Peer review of "DNA methylome and transcriptome analysis established a model of four differentially methylated positions (DMPs) as a diagnostic marker in esophageal adenocarcinoma early detection"

_PeerJ, doi:10.7717/peerj.11355_

## Round 0.1 · original submission · Minor Revisions

Dear Dr. Peng and colleagues:

Thanks for submitting your manuscript to PeerJ. I have now received three independent reviews of your work, and as you will see, the reviewers raised some minor concerns about the research. Despite this, these reviewers are very optimistic about your work and the potential impact it will have on research studying early diagnosis of esophageal adenocarcinoma based on epigenetic markers. Thus, I encourage you to revise your manuscript, accordingly, taking into account all of the concerns raised by the three reviewers.

While the concerns of the reviewers are relatively minor, this is a major revision to ensure that the original reviewers have a chance to evaluate your responses to their concerns. There are many suggestions, which I am sure will greatly improve your manuscript once addressed.

Good luck with your revision,

-joe

Reviewer 1 ·

Basic reporting

The manuscript is well-written and well-organized.

Experimental design

The authors investigated DNA methylation status of the esophageal adenocarcinoma using public databases. Then they developed a robust diagnostic model of this cancer.

Validity of the findings

The developed model was validated by the other public dataset.

Additional comments

The authors should include the name of 4 genes identified here in the abstract.

If possible, the authors should analyze or comment about the differences of methylation patterns between EAC and ESCC. Is this model able to distinguish from EAC from other cancer types?

Reviewer 2 ·

Basic reporting

The work presented by Peng et al. is clear. The manuscript is well-organised and clearly written. The authors made an important effort explaining their results. Moreover, these new findings may serve as a fertile ground for future research in the area of early diagnosis of esophageal adenocarcinoma, based on epigenetic markers.

Experimental design

The experimental methodology is clear to follow.

Validity of the findings

No comment

Additional comments

I have read with interest this research paper. This topic is timely relevant.
Publication is recommended after adressing the following suggestions:
1- Introduction line 66-67: the definition of DNA methylation (.. methylation, a physiological states, involves in methylation of the CpG islands..) could be improved: It is the transfer of a methyl group onto the C5 position of the cytosine..
2- introductuon line 67-68: Although gene silencing by the hypermethylation seems to be the most likely mode of action, there is growing evidence of a more complex view on the effect of DNA methylation in diverse contexts. In particular, for the genes that become hypermethylated, the associated expression level can be unaffected, or even upregulated in some cases (reference: Wan J, Oliver VF, Wang G, Zhu H, Zack DJ, Merbs SL, Qian J. Characterization of
tissue-specific differential DNA methylation suggests distinct modes of positive
and negative gene expression regulation. BMC Genomics. 2015;16:49). The authors could add "in general" after (negatively regulate gene expression), or explain the diversity of epigenetic regulation in this case.
3- methods line 130: stage II/II or I/II ?
4- results line 218: the sentence could be reformulated.

·

Basic reporting

Peng et al report a novel DNA methylome diagnostic signature in patients with Esophageal adenocarcinoma. The authors use a comprehensive tool of plethora of bioinformatic resources to reach some solid conclusions. The writing and the question is clear, with sufficient use of the existing literature. The preparation of the figure is adequate and clear.
I find the topic and the data very interesting and I would support the publication if the authors address some minor, but yet important, comments.

Experimental design

The experimental approach is accurate, with extensive information on the materials and methods section. The scientific question is well-defined and significantly covered throughout the report with high standard analyses.

Validity of the findings

The authors provide their raw data and the statistical model that they have used is efficient and correct. Although their conclusions are well-stated, I would like to suggest some additional analysis in order to straighten their report.

1. In figure 2, the authors should include Gene Ontology analysis of their targets in order to identify potential cancer-related pathways that are affected.

2. Also, the authors should also include Kaplan Meier curves for their proposed 4 gene-signature marker model in patients with Esophageal adenocarcinoma.

3. The authors should investigate and discuss any potential correlation with known mediators of the CpG methylation such as components of the PRC complex.

Additional comments

Peng et al report a novel methylome signature as a diagnostic marker in patients with EA. I believe that this report contributes to our knowledge for the pathophysiology of esophageal adenocarcinoma, and with minor edits this interesting concept should be published. I would be happy to review this work again after their revisions.

---

## Round 0.2 · accepted · Accept

Dear Dr. Peng and colleagues:

Thanks for revising your manuscript based on the concerns raised by the reviewers. I now believe that your manuscript is suitable for publication. Congratulations! I look forward to seeing this work in print, and I anticipate it being an important resource for groups studying early diagnosis of esophageal adenocarcinoma based on epigenetic markers. Thanks again for choosing PeerJ to publish such important work.

Best,

-joe

Reviewer 1 ·

Basic reporting

no comment

Experimental design

no comment

Validity of the findings

no comment

Additional comments

The authors properly revised the manuscript based on our comments.

Reviewer 2 ·

Basic reporting

No comment

Experimental design

No comment

Validity of the findings

No comment

Additional comments

No comment

·

Basic reporting

This is a revised version of Peng et al.
The authors addressed all the comments and suggestions.
I understand that the data of the survival of these markers are not so clinically impactful.
Nevertheless, this concept is worthy publication.

Experimental design

The new data are well controlled and designed.

Validity of the findings

Although the clinical significance of the proposed markers is not strong, it is important that they are discussed and included.

Additional comments

The authors have submitted a significantly improved version of their manuscript.
I support the publication of this interesting concept.